# HIV self-test performance among female sex workers in Kampala, Uganda: a cross-sectional study

Katrina F Ortblad,[1] Daniel Kibuuka Musoke,[2] Thomson Ngabirano,[3] Aidah Nakitende,[2] Geoffrey Taasi,[4] Leah G Barresi,[5] Till Bärnighausen,[6,7,8] Catherine E Oldenburg[9,10,11]

For numbered affiliations see end of article.

**Correspondence to**
Dr Katrina F Ortblad;
katort@uw.edu

## ABSTRACT

**Objective** To evaluate HIV self-testing performance and results interpretation among female sex workers (FSWs) in Kampala, Uganda, who performed unassisted HIV self-testing.

**Methods** In October 2016, 104 participants used an oral HIV self-test while under observation by research assistants. Participants were not assisted on HIV self-test use prior to or during testing, and were only given the manufacturer's pictorial and written instructions to guide them. Research assistants recorded if participants completed and/or had difficulties with steps in the HIV self-testing process on a prespecified checklist. Randomly drawn, used HIV self-tests were interpreted by FSWs. We calculated the concordance between FSWs' interpretations of self-test results with those indicated in the manufacturer's instructions.

**Results** Only 33% (34/104) of participants completed all of the key steps in the HIV self-testing process, and the majority (86%, 89/104) were observed having difficulties with at least one of these steps. Misinterpretation of HIV self-test results were common among FSWs: 23% (12/56) of FSWs interpreted HIV-negative self-test results as HIV positive and 8% (3/37) of FSWs interpreted HIV-positive self-test results as HIV negative. The concordance between FSWs' interpretations of self-test results and that indicated in the instructions was 73% (95% CI 56% to 86%) for HIV-positive self-tests and 68% (95% CI 54% to 80%) for HIV-negative self-tests.

**Conclusions** FSWs in Kampala, who performed unassisted HIV self-testing, skipped steps in the HIV self-testing process and had difficulties correctly interpreting self-test results. Training on use and interpretation of HIV self-tests may be necessary to prevent errors in the HIV self-testing process and to avoid the negative consequences of false-positive and false-negative HIV self-test results among FSWs.

**Trial registration number** NCT02846402.

## Strengths and limitations of this study

► This is one of few studies to explore unassisted oral HIV self-test performance among female sex workers which represents a scenario where female sex workers might obtain HIV self-tests from pharmacies or friends.

► Unlike other studies, female sex workers in this study did not interpret their own HIV self-test result, thus their interpretations were not biased by knowledge of previous HIV test results or HIV risk encounters.

► Study limitations include our small sample size and our limited generalisability of study results to other populations of female sex workers.

## INTRODUCTION

HIV self-testing has been shown to increase HIV testing compared with standard of care HIV testing services in diverse populations[1–8]; however, the benefits of HIV self-testing depend on the self-testers ability to correctly follow a sequence of steps, interpret self-test results and know how to link to appropriate HIV prevention and treatment services. HIV self-testing might be particularly beneficial for female sex workers (FSWs), who are recommended to test for HIV frequently by the WHO.[9] A recent randomised controlled health systems trial found that HIV self-testing significantly increased recent and repeat HIV testing compared with referral to standard HIV testing services among FSWs in Kampala, Uganda.[8]

Previous HIV self-test performance studies found high sensitivity and specificity of participant-interpreted HIV self-test results.[10–19] In the majority of these studies, however, participants were provided with extensive assistance prior the HIV self-testing.[10–12 14 17] For example, in one study participants were asked to demonstrate their understanding of self-testing with a cotton bud and phial of water,[10] while in another study, fisherfolk received a 10 minute demonstration on how to use the self-test prior to testing.[12] In a real-world setting, this level of assistance would be difficult to implement with declining national HIV budgets and other

competing health priorities.[20] In previous HIV self-test performance studies, participants interpreted their own HIV self-test result, potentially introducing bias attributable to knowledge of prior HIV test results and HIV risk encounters.[10–18] Understanding how well individuals can interpret HIV self-test results without bias of prior knowledge is important because as HIV self-testing moves HIV testing outside the unregulated environment of the health system,[8] individuals might use an HIV self-test for first time HIV testing. Few studies to date have measured HIV self-test performance among FSWs,[18 21] an important population for HIV prevention interventions.[22 23]

In this study, we measured unassisted HIV self-test performance (ie, performance in the absence of any assistance prior to or during HIV self-testing) among FSWs in Kampala, Uganda. Specifically, we explore the feasibility and usability of HIV self-tests among FSWs, including FSWs' ability to complete the sequence of steps necessary for self-testing, interpret HIV self-test results, and identify the next steps for HIV treatment and prevention. Additionally, we explore FSWs' values and preferences for HIV self-test, including acceptability and willingness to pay for HIV self-testing. The results of this study are intended to inform implementation of HIV self-testing among members of a key population during a time when few sub-Saharan African countries have guidelines on HIV self-testing.[24]

## METHODS
### Setting
Kampala, the capital city of Uganda, has roughly 13 000 FSWs operating in more than 180 venues. One in every three of these FSWs is estimated to be living with HIV.[25 26] The Ugandan Ministry of Health identifies FSWs as a priority population for HIV prevention interventions[27] and provides them with specialised health services (including clinics and community-based HIV testing) through the Most at Risk Populations Initiative (MARPI).

### Participants
The participants in this study were in training to be peer educators for an HIV self-testing randomised controlled health systems trial among FSWs (Clinical-Trials.gov: NCT02846402).[8] We determined the number of peer educators necessary (120) based on our sample and cluster size for the main trial.[8] To recruit FSW peer educators for the trial, we used established FSW peer organisations in Kampala and MARPI team leaders. Peer educators were eligible for participation if they were 18 years or older and were accepted among members of the FSW community. The peer educator training was 2 days: day 1 covered trial procedures, information on the role of the peer educator, and HIV and sexually transmitted infection prevention, while day 2 covered FSW-friendly HIV testing and treatment options in Kampala and oral HIV self-testing (including how to perform the HIV self-test and interpret the test results). This study took place at

the end of training day 1, prior to any demonstrations of HIV self-testing. Participation in this study was voluntary, and participants were compensated 16 500 Ugandan shillings (UGX) (US$4.70) for their time.

### Usability of HIV self-tests
The participants were silently observed HIV self-testing by research assistants. All participants used the OraQuick ADVANCE Rapid HIV-1/2 Antibody Test (OraSure Technologies, Bethlehem PA), an oral HIV self-test that delivers results in 20 min. Testing took place in large, open rooms that were being used for peer educator training. Three to four participants were in a room at a time and spread out (approximately 10 m apart) so they could not see or hear one another. The participants received no description of the HIV self-testing process or explanation of how to interpret the results prior to testing and were instructed to not ask the research assistant any questions. All participants had access to written and pictorial OraQuick HIV self-test instructions (available in both English and the local language, Luganda) which included information on how to use the self-test as well as information on how to interpret the self-test results and link to care. We did not adapt these instructions for use in this study.

Research assistants recorded the steps participants took when using the HIV self-testing on a standardised checklist, based on the OraQuick instructions. The checklist included: (1) removed buffer cap, (2) put buffer in buffer-stand, (3) swabbed upper or lower gum and (4) put test stick in buffer. Research assistants were instructed to indicate if participants had difficulties with any of these HIV self-testing steps. Additionally, research assistants recorded if participants read the HIV self-test instructions and when they read these instructions. Only one research assistant observed each participant as they HIV self-tested. All used HIV self-tests were taken to a back room (where no participants were present) by the study coordinator so that the results could process in private.

### Interpretation of HIV self-test results
Once the HIV self-test results had processed, these self-tests (including HIV-negative, HIV-positive and invalid results) were placed in a large opaque bag by the study coordinator. Research assistants drew a random HIV self-test from the opaque bag for participants to interpret. Participants only interpreted one randomly drawn HIV self-test instead of many so that previous interpretations of HIV self-test results would not bias later interpretations of HIV self-test results. All valid OraQuick results have at least one red band indicating the control. According to the OraQuick HIV self-testing instructions, a single band indicates a non-reactive or negative test, two red bands indicate a reactive or positive test, and no red bands, blurred bands or no control band indicate an invalid test. Research assistants recorded the results of the HIV self-test they selected by drawing one line, two lines, no lines or blurred lines on a standardised paper form before

handing the HIV self-test to a participant for interpretation. All participants received the manufacturer's written and pictorial instructions to aid their interpretations.

None of the participants in this study learnt their HIV status, as a result study participation and all self-tests were disposed of after study completion. The participants were not given their own self-test to interpret because many already knew their HIV status, and this knowledge would bias our measurement of HIV self-test performance. We were additionally concerned about confidentiality of our participants in a group training environment. Participants who wanted to know their status were referred to clinics in Kampala where they could access free HIV testing and counselling services.

### Perceptions of HIV self-testing

All participants completed a brief questionnaire after interpreting a random HIV self-test result. This questionnaire asked participants to describe the appropriate next steps in HIV treatment and prevention following hypothetical self-test results. The questionnaire also asked participants to specify their belief in the accuracy of the self-test results, interest in future self-testing and willingness to pay for HIV self-testing.

### Statistical analysis

We calculated the proportion of participants that completed steps in the HIV self-testing process, and the proportion of participants who had difficulties with each of these steps. We also used proportions to calculate how participants interpreted HIV-positive, HIV-negative and invalid HIV self-test results. We derived 'correct' interpretations of self-test results using the self-test manufacturer instructions to interpret researcher assistant drawings of the self-test result given to participants. We additionally calculated concordance between participants' interpretations and the manufacturer's interpretations of the strong HIV-positive (two distinct bands) and strong HIV-negative (one distinct band) self-test results. We excluded invalid HIV self-test and weak HIV-positive self-test results from our concordance measurements because these testing outcomes are less common, and we wanted to focus on how well participants could interpret clear results. We did not compare FSW-interpreted HIV self-test results with laboratory blood-based HIV test results because when interpreted correctly by trained users, oral HIV self-testing has >99% sensitivity and specificity.[28–30] We used Stata V.13.1 to estimate binomial 95% confidence intervals (CIs) for our concordance measurements. We then calculated the proportion of participants who described their various perceptions of HIV self-testing, including the next steps following HIV self-test results and accuracy of the HIV self-tests.

### Participant and public involvement

The research question explored in this study was developed from conversations we had with the Ugandan Ministry of Health and was intended to address one of

**Table 1** Sociodemographic characteristics of study participants

| Characteristics | n (%) |
|---|---|
| Age (median, IQR) | 33 (29–37) |
| Education | |
| No formal (0 years) | 4 (4) |
| Primary/junior (<9 years) | 28 (27) |
| Secondary (9–12 years) | 61 (59) |
| Vocational | 5 (5) |
| Tertiary | 6 (6) |
| Can read and write | 94 (90) |
| Previously worked as an FSW peer educator | 86 (83) |
| Times works as a peer educator (median, IQR) | 2 (1–4) |

Sample size, n=104.
FSW, female sex worker.

their concerns related to scaling HIV self-testing. A number of participants were additionally leaders of FSW peer organisations in Kampala and were consulted during the development of the study to gauge their interest in HIV self-testing. These leaders of FSW peer organisations also helped recruitment of study participants and were invited to attend a national dissemination of study results.

### RESULTS

In October 2016, 118 FSWs in urban Kampala, Uganda, completed day 1 of the 2-day peer educator training. Of these, 104 (88%) agreed to perform an HIV self-test, and 99 (84%) interpreted an HIV self-test and completed the questionnaire. Fourteen of the FSWs who participated in day 1 of the 2-day peer educator training refused to perform HIV self-testing, and five of the participants who performed HIV self-testing refused to interpret an HIV self-test result.

Table 1 describes sociodemographic characteristics of study participants. The median age of study participants was 33 years (IQR: 29–27 years). The majority of participants (59%) reported completing some level of secondary education (9–12 years) and 90% of participants reported the ability to read and write. In Uganda, all public schooling on reading and writing is conducted in English. Almost all participants (83%) had worked as a peer educator at least once before participating in this study, 59% of which had previously worked for Uganda's MARPI which we used to help recruit peer educators for this study.

Table 2 highlights the percentage of participants who completed and had difficulties with steps in the HIV self-testing process. While the majority of participants completed each of the individual steps (80% removed the buffer cap, 50% put the buffer in the buffer stand, 80% swabbed the upper or lower gum, 74% put the test stick in the buffer), only 33% of participants completed all four of these steps and the majority of participants (86%)

**Table 2** The percentage of participants who completed and had difficulties with steps in the HIV self-testing process

| Step | Completed n (%) | Had difficulty n (%) |
|---|---|---|
| 1. Removed buffer cap | 80 (80) | 39 (38) |
| 2. Put buffer in buffer-stand | 52 (50) | 72 (69) |
| 3. Swabbed upper or lower gum | 82 (80) | 53 (51) |
| 4. Put test stick in buffer | 77 (74) | 35 (34) |
| All steps 1–4* | 34 (33) | 14 (13) |
| Any step 1–4† | 102 (98) | 89 (86) |
| Read instructions prior to testing | 61 (58) | 51 (49)‡ |
| Referenced instructions during testing | 59 (57) | |

Sample size, n=104. Denominators vary based on completeness in reporting.
*Participants were observed completing all steps 1–4 or having difficulties with all steps 1–4.
†Participants were observed completing any of the steps 1–4 or having difficulties with any of the steps 1-4.
‡Participants were observed having any difficulties with reading the instructions.

were observed by research assistants as having difficulties with at least one of the steps. Of the 89 participants who had difficulties with at least one of the steps, 66% (n=69) were unable to complete at least one of the four indicated steps in the self-testing process. The majority of participants read the manufacturer's self-test instructions prior to testing (58%) or during the HIV self-testing process (57%), but 49% were observed as having difficulties with the instructions.

Of 99 used HIV self-tests that were randomly distributed to participants for self-interpretation, 93 (94%) tests had valid results and 6 (6%) tests had invalid results (four had no result bands and two had no control band but a single test band). Of the valid tests, 56 (57%) had one control band and were considered HIV negative, while 37 (37%)

had both a control and test band and were considered HIV positive (five of these had a weak test band and one had a weak control band).

Figure 1 illustrates how participants interpreted the HIV self-test they were distributed, by how the self-test should have been interpreted according to the OraQuick instructions. Of the 56 HIV-negative self-test results, participants correctly interpreted 38 (68%) as HIV negative and incorrectly interpreted 13 (23%) as HIV positive or invalid. Of the 37 HIV-positive self-test results, participants correctly interpreted 27 (73%) as HIV positive and incorrectly interpreted three as HIV negative (8%). Only one of the invalid self-test results was correctly interpreted as such by participants, the rest were incorrectly interpreted by participants as HIV negative (83%). The concordance between participants' interpretations of self-test results and that indicated in the manufacturer's instructions was 73% (95% CI 56% to 86%) for HIV-positive self-tests and 68% (95% CI 54% to 80%) for HIV-negative self-tests.

When participants were asked what the appropriate next steps for HIV treatment and prevention were following a specified HIV self-test result, few provided the responses suggested in the OraQuick instructions (Table 3). For example, only 25% (24/99) of participants said they would test again in 3 months following an HIV-negative test result, only 46% (45/99) of participants said they would get a confirmatory test to clinic following an HIV-positive test result, and only 58% (51/99) and 36% (32/99) of participants, respectively, said they would retake another oral HIV self-test or get confirmatory test at clinic following an invalid test result.

Figure 2 shows that the majority of participants believed the HIV self-test results, were interested in HIV self-testing in the future and were willing to pay for HIV self-testing. The price that the majority of participants (54%, 53/99) were willing to pay for an HIV self-test was 1000–5000 UGX (US$0.3–US$1.4); 23% (23/99) of participants reported a willingness to pay 5000–10 000 UGX (US$1.4–US$2.88) for an HIV self-test, and only 5% (5/99) of participants

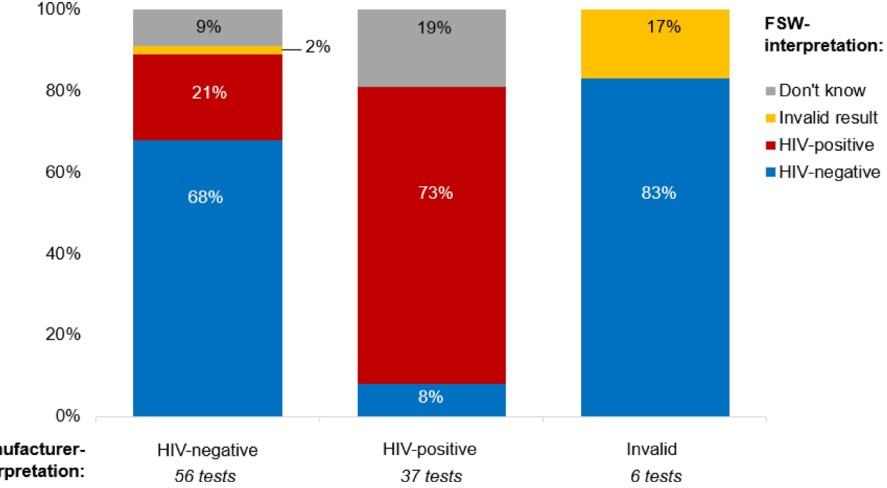

**Figure 1** Participants' interpretations of the HIV self-test results. FSW, female sex worker.

**Table 3** Participant perceptions of the next steps following HIV self-test results. n=99

| Next step | HIV self-test result | | |
| | HIV-negative | HIV-positive | Invalid |
| --- | --- | --- | --- |
| | n (%) | n (%) | n (%) |
| Test again in 12 months* | 0 | 0 | 0 |
| Test again in 3 months | 24 (25) | 3 (3) | 4 (5) |
| Retake another oral HIV self-test | 8 (8) | 7 (7) | 51 (58) |
| Get confirmatory test at clinic | 20 (21) | 45 (46) | 32 (36) |
| Don't know | 2 (2) | 4 (4) | 3 (3) |
| Other: get counselling | – | 21 (21) | 1 (1) |
| Other: start treatment | 1 (1) | 41 (42) | 2 (2) |
| Other: use protection | 53 (53) | 4 (4) | 3 (3) |
| Other: look after yourself | 1 (1) | – | – |
| Other: suicidal thoughts | – | 1 (1) | – |
| Other: test again in 1.5 months | – | 1 (1) | – |
| Other: consult others | – | – | 2 (2) |
| Other: give up on testing | – | – | 1 (1) |

Sample size, n=99.
 (–) participants did not mention this 'Other' category.
*None of the participants responded that they should test for HIV again in 12 months, as indicated with n=0.

reported a willingness to pay greater than 10 000 UGX (US$2.88) for an HIV self-test.

## DISCUSSION

FSWs in urban Uganda, who performed unassisted HIV self-testing, had difficulties completing the steps in the oral HIV self-testing process, correctly interpreting self-test results and understanding the next appropriate steps for HIV treatment and prevention following self-testing. The concordance between FSWs' interpretations and the manufacturer's interpretations of strong HIV-positive and strong HIV-negative self-test results in this study is significantly lower than HIV self-test sensitivity and specificity measurements from other studies,[10–19] but consistent with that measured in another Kampala-based FSW population.[21] Despite challenges using and interpreting the HIV self-tests, the majority of FSWs in this study believed the self-test results and were interested and willing to pay for HIV self-tests in the future.

The low concordance between FSWs' interpretations and the manufacturer's interpretations of strong HIV-positive and strong HIV-negative self-test results and the low percentage of participants that correctly identify the next steps for HIV treatment and prevention following HIV self-testing in this study are concerning. False perception of HIV status, attributable to HIV self-testing may result in behaviours or contribute to mental states among FSWs that are counterproductive to health. For example, an incorrectly perceived HIV-positive self-test result may result in depression,[31] while an incorrectly perceived HIV-negative self-test result may delay HIV treatment. Then, even if FSWs correctly learn their HIV status through self-testing, the inability of participants in this study to correctly identify the next steps in HIV treatment and prevention following HIV self-testing suggests that HIV self-testing may result in missed opportunities to improve individuals' health and prevent HIV transmission.[32 33]

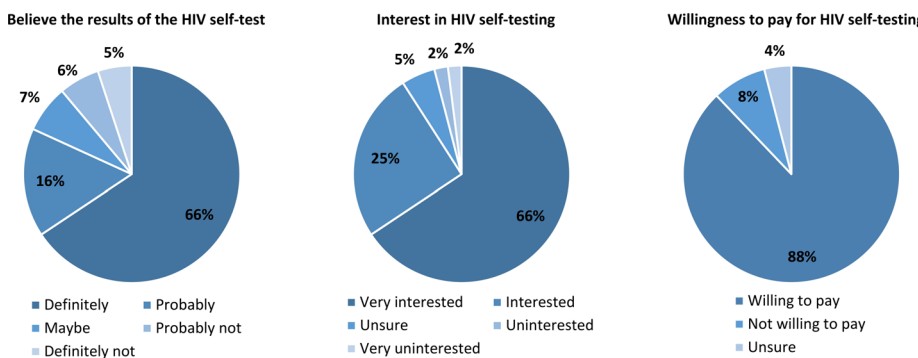

**Figure 2** Participants' belief in the HIV self-test results, interest in HIV self-testing and willingness to pay for HIV self-testing.

The low concordance between FSWs' interpretations and the manufacturer's interpretations of self-test results in this study may be explained by the lack of HIV self-testing assistance provided to study participants. Understanding unassisted HIV self-test performance is important because this reflects a scenario where FSWs may purchase an HIV self-test from a pharmacy or receive one from a friend. Assisted HIV self-testing is also expensive, especially the intensity of those provided in the other HIV self-testing performance studies,[10–17] and thus difficult to implement at scale in the context of limited health budgets and competing HIV prevention interventions.[20] More research should be conducted to determine the appropriate level of assistance, either prior to or during the self-testing process, to achieve high HIV self-test performance.

This study had a number of strengths and limitations. An important strength of this study is that participants interpreted an HIV self-test result that was not their own and thus interpretations of self-test results were not biased by knowledge of previous HIV test results and past HIV risk encounters. Another strength was that participants performed unassisted HIV self-testing which enables us to understand HIV self-testing performance among FSWs in environments where assistance might be unavailable. Limitations of this study included the sample size, which was small because this study was a substudy of a larger HIV self-testing randomised controlled trial.[8] In this study, participants were also observed using the HIV self-tests instead of testing in isolation which may have either positively or negatively influenced study results depending on if this made participants more nervous or careful while self-testing or interpreting self-test results.[12] While unlikely, research assistants many have incorrectly drawn the results of the random HIV self-tests given to participants for interpretation, potentially resulting in a downward bias of our concordance measurements. Some of the self-tests interpreted by study participants may have been interpreted 40 minutes after use, which is outside of the recommended interpretation period. This many have resulted in a greater number of weak HIV-positive self-test results, which are more difficult to interpret. For this reason, weak HIV-positive self-test results were excluded from our concordance measurements. Finally, some participants may have been living with HIV and on antiretroviral treatment, which may have biased their perceptions of the next steps following an HIV-positive self-test result and likely resulted in an overestimation of participants reporting that individuals who self-test HIV-positive should start treatment.

The generalisability of the study results may additionally be limited. First, participants in our study used oral HIV self-tests and thus the results from this study do not represent all forms of HIV self-testing, such as blood-based HIV self-tests. Second, participants were instructed to not ask research assistants for assistance with the HIV self-testing process or interpreting the HIV self-test results which might not represent a scenario where individuals may seek help with self-testing from friends, sexual partners or others. Third, the participants in this study were recommended by leaders within the Kampala FSW community and not necessarily representative of the greater Kampala-based FSW population. The FSW peer educators in this study may have higher health literacy and education compared with other FSWs, potentially resulting in greater HIV self-test performance. Our study results may also have limited generalisability among other populations of FSWs, such as those working in rural setting or transit towns, or those who may not identify as FSWs (ie, barmaids who exchange sex with patrons or young women with 'sugar daddies').[22 23]

## CONCLUSIONS

This study demonstrated that FSWs in urban Uganda have difficulties with the HIV self-testing process and have difficulties interpreting HIV self-test results when unassisted. The findings from this study suggest that governments considering HIV self-testing need to carefully consider what information FSWs are given before they self-test and what resources will be available to them during or after self-testing to prevent mistakes in the self-testing process, misperceptions about HIV status and delays in HIV treatment or prevention interventions. The intensity of HIV self-testing assistance in previous performance studies may be difficult to scale and might not fit within existing national HIV budgets.[20] Simpler HIV self-testing support tools (ie, instructions or demonstration videos), redesign of HIV self-tests or the use of HIV self-testing counsellors should be explored and additional research should be conducted to determine the appropriate level of HIV self-testing assistance.

**Author affiliations**
[1]Department of Global Health, University of Washington, Seattle, WA, USA
[2]International Research Consortium, Kampala, Uganda
[3]Uganda Health Marketing Group, Kampala, Uganda
[4]Uganda Ministry of Health, Kampala, Uganda
[5]Department of Epidemiology, Harvard T.H. Chan School of Public Health, Boston, Massachusetts, USA
[6]Department of Global Health and Population, Harvard T.H. Chan School of Public Health, Boston, Massachusetts, USA
[7]Institute of Public Health, Faculty of Medicine, Heidelberg University, Heidelberg, Germany
[8]Africa Health Research Institute, KwaZulu-Natal, South Africa
[9]Francis I. Proctor Foundation, University of California, San Francisco, California, USA
[10]Department of Ophthalmology, University of California, San Francisco, California, USA
[11]Department of Epidemiology and Biostatistics, University of California, San Francisco, California, USA

**Acknowledgements** We would like to acknowledge all the FSWs who took time participating in this study as well as the research assistants who took time collecting this data.

**Contributors** KFO, DKM and CEO designed the study and conceptualised the paper. KFO conducted the analysis and wrote the first draft. KFO, CEO, TN, DKM, GT, LGB and AN oversaw the collection and the quality data. CEO and TB supervised the study. KFO, DKM, TN, AN, GT, LGB, TB and CEO all edited and revised the manuscript.

**Funding**  This study was funded by the International Initiative for Impact Evaluation (3ie). KFO was additionally supported by the National Institute of Allergy and Infectious Disease T32-AI007535 (PI: Seage) and by the National Institute of Mental Health R01-MH110296 (PI: Heffron) and R01-MH113572 (PIs: Baeten/Ngure). CEO was supported in part by the National Institute on Drug Abuse T32-DA013911 (PI: Flanigan) and the National Institute of Allergy and Mental Health R25-MH083620 (PI: Nunn). TB was funded by the Alexander von Humboldt Foundation through the Alexander von Humboldt Professorship endowed by the German Federal Ministry of Education and Research. He was also supported by the Wellcome Trust, the European Commission, the Clinton Health Access Initiative and NICHD of NIH R01-HD084233, NIAID of NIH R01-AI124389 and R01-AI112339 and FIC of NIH D43-TW009775.

**Disclaimer**  The funders had no role in study design, data collection and analysis, decision to publish or preparation of the manuscript.

**Competing interests**  None declared.

**Patient consent**  Not required.

**Ethics approval**  The Institutional Review Boards at the Harvard T.H. Chan School of Public Health and the Mildmay Uganda Research Ethics Committee. It also received special permission from the Ugandan Ministry of Health to introduce a new HIV testing technology.

**Provenance and peer review**  Not commissioned; externally peer reviewed.

**Data sharing statement**  Data are available from the authors on request.

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
