## [Reviewer comments · BMJ Open]

This paper was submitted to a another journal from BMJ but declined for publication following peer review. The authors addressed the reviewers' comments and submitted the revised paper to BMJ Open. The paper was subsequently accepted for publication at BMJ Open.

(This paper received three reviews from its previous journal but only two reviewers agreed to published their review.)

ARTICLE DETAILS

TITLE (PROVISIONAL)	HIV self-test performance among female sex workers in Kampala, Uganda: a cross-sectional study
AUTHORS	Ortblad, Katrina F; Kibuuka Musoke, Daniel; Ngabirano, Thomson; Nakitende, Aidah; Taasi, Geoffrey; Barresi, Leah G; Bärnighausen, Till; Oldenburg, Catherine E

VERSION 1 – REVIEW

REVIEWER	Augustine Choko Malawi Liverpool Wellcome Trust Clinical Research Programme, Malawi
REVIEW RETURNED	03-Apr-2018

GENERAL COMMENTS	Title: very informative Key words: suggest you add Africa, Uganda Abstract Line 6: should be objective not objectives? Line 7: ...in the absence of pre-test training. Is this referring to unsupervised HIV self-testing? Line 32: "has" should be "had" Lines 32-33: The conclusion that FSWs had difficulty correctly using self-test kits is not supported by the data. The results show problems with interpretation which does not mean difficulty in using the kits which implies difficulty opening the kit or performing any other procedure prior to reading the results. Main article Page 3 Line 8: "is" is missing between This and the Page 4 Lines 44-50: I feel like in practice people are more careful with their own test results and would pay extra attention compared to being asked a random test. Page 6 Line 21: It is said elsewhere that FSWs have lower literacy levels but here it is reported that all gave written informed consent. Were there any who needed witnessed consent? Line 39: ...instructed not to ask for instructions. In real life people may seek help with testing from friends, partners, relatives etc. This may in turn improve difficulties with the testing process.
---

	Page 7 Line 28: Given that this is a population encouraged to test regularly, not giving them HIV results was a lost opportunity. Referring the FSWs to another clinic may have introduced the distance and other barriers of access. Line 48: "in" is missing between measured and two Page 8 Line 26: 80 of 99 participants removed the buffer cap. This is very interesting because without removing the buffer cap, how really can one perform the test? It means only 80 FSWs could be considered to have performed the self-test. Indeed this shows that only having performed your own testing can you interpret the results. This fact, that other FSWs in fact did not even perform their self-test, was ignored in the distribution of the random self-tests. This could explain the high errors rates in interpreting the self-tests. Results section needs to describe some demographic characteristics of the study participants especially their literacy as this seems to be assumed throughout the study implementation. Page 12 Lines 41-46: No demographic information is give for the research participants but the authors state that better interpretation could be explained by better literacy which is not presented in the manuscript.
--	--

REVIEWER	Carmen Figueroa World Health Organisation, Switzerland
REVIEW RETURNED	05-Apr-2018

GENERAL COMMENTS	Dear authors Congratulations on your effort to prepare this manuscript. Overall I thought this manuscript could be presented in a more nuanced manner. Specific comments to that end are found below. 1.MAJOR COMMENTS Methods 1.Outcomes of the study should be specified and better defined. This study does not have the proper design to evaluate "HIV testing" process; HIV testing should be done using a validated algorithm of at least 2 or 3 different assays. The HIVST "process" should be better explained or use another terminology; does the process includes from opening to proper disposal of the device? Interpretation of results should be a part of this "HIVST process". Please clarify the term "belief" are you referring to confidence/trustfulness of results or the FSW's opinion/conviction about HIVST? 2. Please consider that an HIVST does NOT provide a definitive result, reactive results always require further testing and confirmation using validated national algorithm and non-reactive results will depend on the ongoing risk of HIV exposure. Therefore your approach to evaluate the "accuracy", by RESEARCH ASSISTANT late reading of results should be reconsidered. I believe what you are considering is the concordance between FSW and RESEARCH ASSISTANT. Also was the same RESEARCH ASSISTANT interpreting all the results? Do you have any information on his/her ability to correctly interpret the results? 3. Please consider re-wording "mock-up results", participants are interpreting actual results. Not their own. Please provide more information how the random selection is done, when participants finish where (sealed box) or to who they give the test? If they give the test to RESEARCH ASSISTANT, how the distribution decided? 4. Please revise some information included in the background. The phrase mentioning that previous HIVST studies evaluating
---

performance provided “pre-training” is not accurate, Kurth 2016, Dong 2014 and Napierala 2015 evaluated performance of HIVST without assistance. Also few studies have evaluated the HIVST performance among key populations, including FSW: Marley 2014.

5. Please include a table with socio-demographic characteristics of FSW, you are stating they have low level of education; you should provide the evidence and the references where pertinent.

6. Objective should be clearly specified: explore feasibility and usability of HIVST, describe values and preferences among FSW including: acceptability, willingness to pay for HIVST, etc.

Results

1. Revise the use of confidence intervals; this is not the appropriate statistical analysis. Suggest using an exploratory and descriptive analysis and evaluate statistical significance of differences using the chi2 test for proportions.

Discussion

1. First paragraph should summarize results, related with your findings, your research did not compare groups receiving and not receiving a demonstration before HIVST, how can you relate/state that the absence of assistance can create difficulties when using an HIVST?

2. Suggest to revise your limitations, you should explore the effect of late-reading, the effect of interpreting another person result, self-knowledge of HIV status, awareness of ARV use.

3. Your findings are not applicable to all types/approaches of HIVST, your intervention used an oral based kit and an unassisted intervention.

4. Revise your conclusion; conclusion should be justified by your results. I found not enough evidence to state that absence of pre-training can result in errors when using HIVST.

2. MINOR COMMENTS

1. Abstract and paper could do with another edit. There are repetitive or unclear phrases that needs revision. i.e ethical approval is mentioned in ethical approval section and in result interpretation section.

2. Please revise results and discussion to take in consideration all my comments.

3. Please consider changing the title. Consider following the guidelines of the Publication Manual of the American Psychological Association (APA, 2009). The manual recommends simplicity and the use of concise statements when formulating your title.

Methods

1. Please provide more information how participant selection. Were they recommended? By who? From which areas they were selected.

2. Provide clear information about the two-day training. Not clear to me which participants received the HIVST training or not. You mentioned “this study took place at the end of training day one, prior to any discussion around HIV testing options or demonstration of HIVST”, but then in results you described how many participants were in the two-day training.

3. Please provide more information about the space were FSW performed the test, were they in cubicles or was it an open space, if they were in an open space what was the distance between them?

4. Please revise information about instructions to interpret results, your study is using a specific brand, you should not generalize instructions to interpret reactive, non-reactive or invalids results

5. Please justify why invalids results from FSW were excluded.

6. Please clearly explain how data was analyzed, you estimated binomial 95% CI for which variables. Did you used raw diagnostics (false positive, false negative, true positive, true negative) to

	calculate sensitivity/specificity? Suggest you to calculate concordance, otherwise please justify why not. You are not comparing HIVST results to a “gold standard” sensitivity/specificity are not appropriate measures. 7. Please explain how observation was performed and how many research assistants were observing the FSW. Was the research assistant taking note during or after the process? Also please include this effect in the discussion or the limitations Results 1. How many participants were offered to participate and how many refused? Were there any lost participants? 2. Please do not repeat information in tables and figures. Figure 2 and table in appendix 1 have the same information. 3. Revise titles of figures and tables. The title should clearly indicate what results are shown in the table/figure. Table 1 for example - completed task: tasks accomplished in a defined period of time, the categories are mentioning steps for product use. Please revise CI95% calculation for tables, this statistical analysis does not correspond. CI95% are population parameters for which the difference between the parameter and the observed estimate is not statistically significant. 4. Please provide more information about steps performance: did all the FSW read instructions fully or partially, difficulties removing buffer cap, buffer stand, swabbing their gums and putting the stick in the buffer. It's important to provide nuances about completeness and difficulties in the step. i.e. 77/99 were able to swab upper gum, but how many had difficulties doing so.
--	---

REVIEWER	Enrico Girardi National Institute for Infectious Diseases "L. Spallanzani" IRCCS, Rome, Italy
REVIEW RETURNED	12-Apr-2018

GENERAL COMMENTS	This paper provides useful information for those planning the implementation of HIV serl test program Some more details should be provided Regarding the written information provided to participant, was it a mere translation form English to local language? For both the written and pictorial information, there was any adaption to local context and was understandability for local population pre-tested? Is there any information on literacy/education of participants? Did they understand English?
---

VERSION 1 – AUTHOR RESPONSE

Reviewer: 1

Reviewer Name: Augustine Choko

Institution and Country: Malawi Liverpool Wellcome Trust Clinical Research Programme, Malawi

Please state any competing interests: None declared

Title: very informative

We thank the reviewer for this comment.

Key words: suggest you add Africa, Uganda

We have added the key words “Africa” and “Uganda”.

Abstract

Line 6: should be objective not objectives?

We have changed “Objectives” to “Objective”.

Line 7: ...in the absence of pre-test training. Is this referring to unsupervised HIV self-testing?

While the research assistants in this study did not aid the participants in the HIV self-testing process, they observed them as they used an HIV self-test and interpreted a random HIV self-test result. For this reason, we believe it is misleading to refer to the HIV self-testing in this study as “unsupervised”, and thus have described it as HIV testing in the absence of pre-test counseling. We have clarified this point.

(Abstract, Objective): “To evaluate HIV self-testing performance and results interpretation among female sex workers (FSWs) in Kampala, Uganda who did not receive pre-test training.”

Line 32: “has” should be “had”

We have addressed this error. The sentence now reads as:

(Abstract, Conclusions): “FSWs in Kampala, who did not receive any pre-test training, skipped steps in the HIV self-testing process and had difficulties correctly interpreting self-test results.”

Lines 32-33: The conclusion that FSWs had difficulty correctly using self-test kits is not supported by the data. The results show problems with interpretation which does not mean difficulty in using the kits which implies difficulty opening the kit or performing any other procedure prior to reading the results.

We have clarified in the methods sub-section how we measured how well participants used the HIV self-tests, reported these findings in the results sub-section, and incorporated the significance of these findings in the conclusions sub-section:

(Abstract, Methods): “Research assistants recorded if participants completed and/or had difficulties with steps in the HIV self-testing process on a pre-specified checklist.”

(Abstract, Results): “Only 33% (34/104) of participants completed all of the key steps in the HIV self-testing process and the majority (86%, 89/104) were observed having difficulties with at least one of these steps.”

(Abstract, Discussion): “FSWs in Kampala, who did not receive any pre-test training, skipped steps in the HIV self-testing process and had difficulties correctly interpreting self-test results.”

Main article

Page 3, Line 8: “is” is missing between This and the.

We have corrected this error:

(Article Summary, Bullet 1): “This is one of few studies to explore oral HIV self-test performance among female sex workers in the absence of pre-test training”

Page 4, Lines 44-50: I feel like in practice people are more careful with their own test results and would pay extra attention compared to being asked a random test.

We agree with the reviewer that participants may be more careful when interpreting their own self-test results, but we also believe that when participants interpret their own self-test result they have access to additional information (e.g., knowledge of prior HIV test results and HIV risk encounters) that may bias their interpretation of the self-test result. We discuss the potential bias introduced by having participants interpret their own self-test result and also discuss the potential limitation of not having participants do this in our study.

(Introduction, Paragraph 2): “In previous HIV self-test performance studies participants interpreted their own HIV self-test result, potentially introducing bias attributable to knowledge of prior HIV test results and HIV risk encounters.[14–22] Understanding how well individuals can interpret HIV self-test results without bias of prior knowledge is important because as HIV self-testing moves HIV testing outside the unregulated environment of the health system,[8] individuals might use an HIV self-test to test others (e.g., sexual partners, children) or use an HIV self-test for first time HIV testing.”

(Discussion, Paragraph 5): “Sixth, participants did not interpret their own HIV self-test results and thus may have been less careful when interpreting a random HIV self-test (especially since many of them likely already knew their HIV status). However, we have previously described the advantages of this study design.”

Page 6, Line 21: It is said elsewhere that FSWs have lower literacy levels but here it is reported that all gave written informed consent. Were there any who needed witnessed consent?

In our study, participants who could not provide written informed consent provided consent with a thumbprint. We have clarified this in the manuscript:

(Methods, Participants, Paragraph 1): “All participants provided written informed consent in the language in which they were most comfortable. Those who could not write provided consent with a thumbprint.”

Page 6, Line 39: ...instructed not to ask for instructions. In real life people may seek help with testing from friends, partners, relatives etc. This may in turn improve difficulties with the testing process.

The reviewer brings up an important point. In our discussion we have noted how performing self-testing in the absence of not being able to consult with others might not reflect a real-life scenario:

(Discussion, Paragraph 6): “Second, participants were instructed to not ask research assistants for assistance with the HIV self-testing process or interpreting the HIV self-test results, which might not represent a real-life scenario where individuals may seek help with self-testing from friends, sexual partners, or others.”

Page 7, Line 28: Given that this is a population encouraged to test regularly, not giving them HIV results was a lost opportunity. Referring the FSWs to another clinic may have introduced the distance and other barriers of access.

The reviewer brings up an important point. In the methods section, we have justified why we did not give participants the results of their HIV self-test, which was for study design and confidentiality reasons. This study design was approved by ethics boards in Kampala and Boston.

(Methods, Interpretation of HIV self-test results, Paragraph 2): “None of the participants in this study learned their HIV status as a result study participation and all self-tests were disposed of after study completion. The participants were not given their own self-test to interpret because many already knew their HIV status and this knowledge would bias our measurement of HIV self-test performance. We were additionally concerned about confidentiality of our participants

in a group training environment. Participants who wanted to know their status were referred to clinics in Kampala where they could access free HIV testing and counseling services.”

Page 7, Line 48: “in” is missing between measured and two

We have extensively edited and rearranged the methods section, addressing all grammatical errors.

Page 8, Line 26: 80 of 99 participants removed the buffer cap. This is very interesting because without removing the buffer cap, how really can one perform the test? It means only 80 FSWs could be considered to have performed the self-test. Indeed this shows that only having performed your own testing can you interpret the results. This fact, that other FSWs in fact did not even perform their self-test, was ignored in the distribution of the random self-tests. This could explain the high error rates in interpreting the self-tests.

In this study, we used two different methods to measure self-test performance and interpretation of self-test results that were not dependent upon one another. Since participants interpreted a random HIV self-test result that was not their own, how well participants interpreted an HIV self-test result did not depend on their correct use of the HIV self-test.

It is possible, however, that participants who did not use the HIV self-test (i.e., those who did not remove the buffer cap) felt less confident in their interpretation of self-test results, resulting in higher error rates in interpreting self-test results. We have added this to the discussion section:

(Discussion, Paragraph 4): “Second, only 80% of participants in this study removed the buffer cap of the HIV self-test, enabling them to properly HIV self-test. Those who did not properly use the HIV self-test may have had less confidence in their ability interpret self-test results, resulting in increased errors.”

Results section needs to describe some demographic characteristics of the study participants especially their literacy as this seems to be assumed throughout the study implementation.

We have added Table 1 to the results section, which describes the socio-demographic characteristics of study participants:

Table 1. Socio-demographic characteristics of study participants.

N=104

Characteristics	N (%)
Age (median, IQR)	33 (29 to 37)
Education	
No formal (0 years)	4 (4%)
Primary/Junior (<9 years)	28 (27%)
Secondary (9-12 years)	61 (59%)
Vocational	5 (5%)
Tertiary	6 (6%)
Can read and write	94 (90%)
Previously worked as a FSW peer educator	86 (83%)
Times works as a peer educator (median, IQR)	2 (1 to 4)

This has additionally been described in the text:

(Results, Paragraph 2): “Table 1 describes socio-demographic characteristics of study participants. The median age of study participants was 33 years (interquartile range: 29 to 27 years). The majority of participants (59%) reported completing some level of secondary education (9 to 12 years) and 90% of participants reported the ability to read and write. In Uganda, all public schooling on reading and writing is conducted in English. Almost all participants (83%) had worked as a peer educator at least once before participating in this study, 59% of which had previously worked for Uganda’s Most at Risk Population Initiative (also known as MARPI), which we used to help recruit peer educators for this study.”

Page 12, Lines 41-46: No demographic information is given for the research participants but the authors state that better interpretation could be explained by better literacy which is not presented in the manuscript.

We have described the self-reported education and literacy rates of our study population in the results section:

(Results, Paragraph 1): “The majority of participants (59%) reported completing some level of secondary education (9 to 12 years) and 90% of participants reported the ability to read and write.”

We have discussed how differences in health literacy and education among FSWs compared to members of the general population is an unlikely explanation of low HIV self-test performance in this study:

(Discussion, Paragraph 4): “First, our study focused on FSWs, a key population that we hypothesized may have increased difficulties with HIV self-test performance because of often cited lower levels of health literacy [10] and educational attainment [11–13] compared to members of the general population. Lower levels of health literacy and educational attainment, however, is an unlikely explanation for low self-test performance in this study because the majority of participants self-reported the ability to read and write.”

Reviewer: 2

Reviewer Name: Carmen Figueroa

Institution and Country: World Health Organization, Switzerland

Please state any competing interests: None declared

Dear authors, Congratulations on your effort to prepare this manuscript. Overall I thought this manuscript could be presented in a more nuanced manner. Specific comments to that end are found below.

We thank the reviewer for their comment and have addressed each of their specific comments below.

1.MAJOR COMMENTS

Methods

1.Outcomes of the study should be specified and better defined. This study does not have the proper design to evaluate “HIV testing” process; HIV testing should be done using a validated algorithm of at least 2 or 3 different assays. The HIVST “process” should be better explained or use another terminology; does the process includes from opening to proper disposal of the device? Interpretation of results should be a part of this “HIVST process”. Please clarify the term “belief” are you referring to confidence/trustfulness of results or the FSW’s opinion/conviction about HIVST?

We have extensively edited the methods section of this manuscript to improve clarity of the outcomes we are measuring.

First, we more clearly defined these outcomes at the end of the introduction:

(Introduction, Paragraph 3): “In this study, we measured HIV self-test performance in the absence of any pre-test training among FSWs in Kampala, Uganda. Specifically, we explore the feasibility and usability of HIV self-tests among FSWs, including FSWs’ ability to complete the sequence of steps necessary for self-testing, interpret HIV self-test results, and identify the next steps for HIV treatment and prevention. Additionally, we explore FSWs’ values and preferences for HIV self-test, including acceptability and willingness to pay for HIV self-testing.”

Second, we have broken up our methods section into sub-sections that map to these, including: “usability of HIV self-tests”, “interpretation of HIV self-test results”, and “perceptions of HIV self-testing”. As you can see, we have replaced an evaluation of the “HIV self-testing process” with measurement of “HIV self-test performance”, including usability of HIV self-tests and interpretation of HIV self-test results.

Throughout the text we have clarified that we are measuring participants’ belief in the accuracy of HIV self-test results.

2. Please consider that an HIVST does NOT provide a definitive result, reactive results always require further testing and confirmation using validated national algorithm and non-reactive results will depend on the ongoing risk of HIV exposure. Therefore your approach to evaluate the “accuracy”, by RESEARCH ASSISTANT late reading of results should be reconsidered. I believe what you are considering is the concordance between FSW and RESEARCH ASSISTANT. Also was the same RESEARCH ASSISTANT interpreting all the results? Do you have any information on his/her ability to correctly interpret the results?

We have extensively edited the methods section for clarity. We have changed our language so that we are no longer measuring HIV self-test “accuracy”, but rather participants’ ability to interpret self-test results relative to a research assistant drawing.

Under the statistical analysis sub-section in the methods section we have clarified how we measured FSW-interpreted HIV self-test sensitivity and specificity and our justification for this approach:

(Methods, Statistical analysis): “To measure FSW-interpreted HIV self-test sensitivity and specificity, we calculated the proportion of participants that correctly interpreted the strong HIV-positive (two clear bands) and strong HIV-negative (one clear band) self-test results. The ‘correct interpretation’ of the self-test result was determined from the research assistant drawings of the self-test results, which were interpreted as indicated in the manufacturer’s instruction guide. We excluded invalid HIV self-test and weak HIV-positive self-test results from our FSW-interpreted HIV self-test sensitivity and specificity calculations because these testing outcomes are less common and we wanted to focus on how well participants could interpret clear self-test results. We did not compare FSW-interpreted HIV self-test results with laboratory blood-based HIV test results, because when interpreted correctly by trained users, oral HIV self-testing has >99% sensitivity and specificity.[31–33]”

In our study limitations, we discuss how inaccurate drawings of HIV self-test results by research assistants many have biased our estimates of FSW-interpreted HIV self-test sensitivity and specificity:

(Discussion, Paragraph 5): “Fourth, while unlikely, research assistants many have incorrectly drawn the results of random HIV self-tests given to participants for interpretation, potentially resulting in a downward bias of our FSW-interpreted HIV self-test sensitivity and specificity estimates.”

3. Please consider re-wording “mock-up results”, participants are interpreting actual results. Not their own. Please provide more information how the random selection is done, when participants finish where (sealed box) or to who they give the test? If they give the test to RESEARCH ASSISTANT, how the distribution decided?

We no longer refer to HIV self-test results as “mock-up results” throughout the text.

We have provided more information where HIV self-tests went once they were used by participants and how research assistants selected a random HIV self-test for participants to interpret:

(Methods, Usability of HIV self-tests, Paragraph 2): “All used HIV self-tests were taken to a back room (where no participants were present) by the study coordinator so that the results could process in private.”

(Methods, Interpretation of HIV self-test results, Paragraph 1): “Once the HIV self-test results had processed, these self-tests were placed in a large opaque bag by the study coordinator. Research assistants drew a random HIV self-test from this bag for FSWs to interpret.”

4. Please revise some information included in the background. The phrase mentioning that previous HIVST studies evaluating performance provided “pre-training” is not accurate, Kurth 2016, Dong 2014 and Napierala 2015 evaluated performance of HIVST without assistance. Also few studies have evaluated the HIVST performance among key populations, including FSW: Marley 2014.

We have revised the background section so that we are only referencing the studies that provided pre-test training when discussing this as a limitation of previous studies:

(Introduction, Paragraph 2): “In the majority of these studies, however, participants were provided with extensive pre-test training prior to HIV self-testing.[14–16,18,21]”

We now discuss how this is one of few studies to evaluate HIV self-test performance among female sex workers, an important population for HIV prevention interventions:

(Article Summary): “This is one of few studies to explore oral HIV self-test performance among female sex workers in the absence of pre-test training; a realistic scenario where female sex workers might obtain HIV self-tests from pharmacies or friends.”

(Introduction, Paragraph 2): “Few studies to date has measured HIV self-test performance among FSWs,[22,25] an important population for HIV prevention interventions.[13,26]”

5. Please include a table with socio-demographic characteristics of FSW, you are stating they have low level of education; you should provide the evidence and the references where pertinent.

We have added Table 1, which describes the socio-demographic characteristics of study participants:

Table 1. Socio-demographic characteristics of study participants.

N=104

Characteristics	N (%)
Age (median, IQR)	33 (29 to 37)
Education	
No formal (0 years)	4 (4%)

Primary/Junior (<9 years)	28 (27%)
Secondary (9-12 years)	61 (59%)
Vocational	5 (5%)
Tertiary	6 (6%)
Can read and write	94 (90%)
Previously worked as a FSW peer educator	86 (83%)
Times works as a peer educator (median, IQR)	2 (1 to 4)

We discuss how levels of education and literacy were higher in this population than expected, and thus were an unlikely explanation of low HIV self-test performance in this study:

(Discussion, Paragraph 4): “First, our study focused on FSWs, a key population that we hypothesized may have increased difficulties with HIV self-test performance because of often cited lower levels of health literacy [10] and educational attainment [11–13] compared to members of the general population. Lower levels of health literacy and educational attainment, however, is an unlikely explanation for low self-test performance in this study because the majority of participants self-reported the ability to read and write.”

6. Objective should be clearly specified: explore feasibility and usability of HIVST, describe values and preferences among FSW including: acceptability, willingness to pay for HIVST, etc. Results

We have clarified the objectives of this study in the introduction:

(Introduction, Paragraph 3): “In this study, we measured HIV self-test performance in the absence of any pre-test training among FSWs in Kampala, Uganda. Specifically, we explore the feasibility and usability of HIV self-tests among FSWs, including FSWs’ ability to complete the sequence of steps necessary for self-testing, interpret HIV self-test results, and identify the next steps for HIV treatment and prevention. Additionally, we explore FSWs’ values and preferences for HIV self-test, including acceptability and willingness to pay for HIV self-testing.”

1. Revise the use of confidence intervals; this is not the appropriate statistical analysis. Suggest using an exploratory and descriptive analysis and evaluate statistical significance of differences using the chi2 test for proportions.

We have revised the manuscript so that it is primarily a descriptive analysis where we measure the percentage of participants who report various study outcomes. We describe this in our extensively revised methods section:

(Methods, Statistical analysis, Paragraph 1): “We calculated the proportion of participants that completed steps in the HIV self-testing process and the proportion of participants that had difficulties with each of these steps. To measure FSW-interpreted HIV self-test sensitivity and specificity, we calculated the proportion of participants that correctly interpreted the strong HIV-positive (two clear bands) and strong HIV-negative (one clear band) self-test results. The ‘correct interpretation’ of the self-test result was determined from the research assistant drawings of the self-test results, which were interpreted as indicated in the manufacturer’s instruction guide. We excluded invalid HIV self-test and weak HIV-positive self-test results from our FSW-interpreted HIV self-test sensitivity and specificity calculations because these testing outcomes are less common and we wanted to focus on how well participants could interpret clear self-test results. We did not compare FSW-interpreted HIV self-test results with laboratory blood-based HIV test results, because when interpreted correctly by trained users, oral HIV self-testing has >99% sensitivity and specificity.[31–33] Stata 13.1 (Stata Corporation, College Station, TX) was used to estimate binomial 95% confidence intervals for these

measurements. We then calculated the proportion of participants that described their various perceptions of HIV self-testing, including the next steps following HIV self-test results and accuracy of the HIV self-tests.”

We have justified our calculation of FSW-interpreted HIV self-test sensitivity and specific measurements in the section above and described how we calculate the 95% CIs for these estimates.

Discussion

1. First paragraph should summarize results, related with your findings, your research did not compare groups receiving and not receiving a demonstration before HIVST, how can you relate/state that the absence of assistance can create difficulties when using an HIVST?

We have extensively revised the discussion section so that our conclusions more specifically relate to the study design findings. Throughout the discussion section, we have deemphasized the effect of not having pre-test training on our findings and have instead used this as a descriptor of our study population. For example:

(Discussion, Paragraph 1): “FSWs in urban Uganda, who did not receive pre-test training, had difficulties completing the steps in the oral HIV self-testing process, correctly interpreting self-test results, and understanding the next appropriate steps for HIV treatment and prevention following self-testing.”

2. Suggest to revise your limitations, you should explore the effect of late-reading, the effect of interpreting another person result, self-knowledge of HIV status, awareness of ARV use.

We have extensively revised the limitations of this study at the suggestion of the reviewer. Our limitations now specifically discuss the potential effect of late-reading, interpreting another person’s HIV self-test result, knowledge of HIV status, and awareness of ARV use:

(Discussion, Paragraph 5): “There are limitations with this study that are important to note. First, the sample size was small because this study was a sub-study of a larger HIV self-testing randomized controlled trial.[8] Second, the FSWs in this study were observed using the HIV self-tests instead of testing in isolation, which may have either positively or negatively influenced study results depending on if this made FSWs more nervous or careful while self-testing or interpreting self-test results.[16] Third, the FSWs in this study were instructed to not ask research assistants for assistance with the HIV self-testing process or interpreting the HIV self-test results, which might not represent a real life scenario where individuals may seek help with self-testing from friends, partner or others. Fourth, while unlikely, research assistants many have incorrectly drawn the results of random HIV self-tests given to participants for interpretation, potentially resulting in a downward bias of our FSW-interpreted HIV self-test sensitivity and specificity estimates. Fifth, some of the self-tests interpreted by study participants may have been interpreted 40 minutes after use, which is outside of the recommended interpretation period. This many have resulted in a greater number of weak HIV-positive self-test results, which are more difficult to interpret. For this reason, weak HIV-positive self-test results were excluded from our measurement of FSW-interpreted HIV self-test sensitivity. Sixth, participants did not interpret their own HIV self-test results and thus may have been less careful when interpreting a random HIV self-test (especially since many of them likely already knew their HIV status). However, we have previously described the advantages of this study design. Finally, some participants may have been living with HIV and on antiretroviral treatment, which may have biased their perceptions of the next steps following an HIV-positive self-test result and likely resulted in an overestimation of participants reporting that individuals who self-test HIV-positive should start treatment.”

3. Your findings are not applicable to all types/approaches of HIVST, your intervention used an oral based kit and an unassisted intervention.

The reviewer brings up an important point, we have added this as one of the limitations to the generalizability of our study results:

(Discussion, Paragraph 6): “The generalizability of the study results may additionally be limited. First, participants in our study used oral HIV self-tests and thus the results from this study do not represent all forms of HIV self-testing, such as blood-based HIV self-tests.”

4. Revise your conclusion; conclusion should be justified by your results. I found not enough evidence to state that absence of pre-training can result in errors when using HIVST.

We have extensively revised our conclusions so that they are now justified by the results. Like the rest of the discussion, we have deemphasized the effect of not providing participants with pre-test training.

(Conclusions): “The findings from this study suggest that governments considering HIV self-testing need to carefully consider what information FSWs are given before they self-test and what resources will be available to them during or after self-testing to prevent mistakes in the self-testing process, misperceptions about HIV status, and delays in HIV treatment or prevention interventions. The intensity of pre-test training in previous HIV self-test performance studies may be difficult to scale and might not fit within existing national HIV budgets.[24] Simpler HIV self-testing support tools (i.e., instruction guides or demonstration videos), redesign of HIV self-test, or the use of HIV self-testing counselors should be explored and additional research, including randomized trials, should be conducted to determine the appropriate level of pre-test training.”

2. MINOR COMMENTS

1. Abstract and paper could do with another edit. There are repetitive or unclear phrases that needs revision. i.e ethical approval is mentioned in ethical approval section and in result interpretation section.

We have extensively edited the abstract and paper to improve clarity. We now only mention ethical approval under the ethics sub-section of the methods section:

(Methods, Ethics, Paragraph 1): “The study was approved by the Institutional Review Boards at the Harvard T.H. Chan School of Public Health and the Mildmay Uganda Research Ethics Committee, it also received special permission from the Ugandan Ministry of Health to introduce a new HIV testing technology.”

2. Please revise results and discussion to take in consideration all my comments.

The results and discussion section of this manuscript have been extensively revised to address all reviewer comments.

3. Please consider changing the title. Consider following the guidelines of the Publication Manual of the American Psychological Association (APA, 2009). The manual recommends simplicity and the use of concise statements when formulating your title.

We have simplified the title of our manuscript to: “HIV self-test performance among female sex workers in Kampala, Uganda: a cross-sectional study”.

Methods

1. Please provide more information how participant selection. Were they recommended? By who? From which areas they were selected.

We have added more information about how participants were selected for this study under the participants sub-section of the methods section:

(Methods, Participants, Paragraph 1): “The participants in this study were in training to be peer educators for an HIV self-testing randomized controlled health systems trial among FSWs.[8] We determined the number of peer educators necessary (120) based on our sample and cluster size for the main trial.[8] To recruit FSW peer educators for the trial, we used established FSW peer organizations in Kampala and MARPI team leaders. Peer educators were eligible for participation if they were 18 years or older and were accepted among members of the FSW community.”

2. Provide clear information about the two-day training. Not clear to me which participants received the HIVST training or not. You mentioned “this study took place at the end of training day one, prior to any discussion around HIV testing options or demonstration of HIVST”, but then in results you described how many participants were in the two-day training.

None of the participants in this study received HIV self-test training prior to study participation. We have clarified our description of the two-day peer educator training in our methods section:

(Methods, Participants, Paragraph 1): “The participants in this study were in training to be peer educators for an HIV self-testing randomized controlled health systems trial among FSWs.[8] ... The peer educator training was two days: day one covered trial procedures, information on the role of the peer educator, and HIV and STIs prevention, while day two covered FSW-friendly HIV testing and treatment options in Kampala and oral HIV self-testing (including how to perform the HIV self-test and interpret the test results). This study took place at the end of training day one, prior to any demonstrations of HIV self-testing.

In our results section, we have clarified that we are reporting outcomes for participants who completed day one of the two-day peer educator training:

(Results, Paragraph 1): “In October 2016, 118 FSWs in urban Kampala, Uganda completed day one of the two-day peer educator training. Of these, 104 (88%) agreed to perform an HIV self-test and 99 (84%) interpreted a HIV self-test and completed the questionnaire.”

3. Please provide more information about the space where FSW performed the test, were they in cubicles or was it an open space, if they were in an open space what was the distance between them?

We have elaborated on the space where FSWs performed the HIV self-test in the methods section:

(Methods, Usability of HIV self-tests, Paragraph 1): “Testing took place in large, open rooms that were being used for peer educator training. Three to four FSWs were in a room at a time and spread out (approximately 10 meters apart) so they could not see or hear one another.”

4. Please revise information about instructions to interpret results, your study is using a specific brand, you should not generalize instructions to interpret reactive, non-reactive or invalid results.

In our description of how to interpret HIV self-test results, we have clarified that this interpretation is for the OraQuick HIV self-tests (i.e., those used in this study):

(Methods, Interpretation of HIV self-test results, Paragraph 1): “All valid OraQuick results have at least one red band indicating the control. According to the OraQuick HIV self-testing instruction guide, a single band indicates a nonreactive or negative test, two red bands

indicate a reactive or positive test, and no red bands, blurred bands, or no control band indicate an invalid test.”

5. Please justify why invalid results from FSW were excluded.

We have justified why invalid results were excluded from our measurements of FSW-interested HIV self-test sensitivity and specificity in our methods section:

(Methods, Statistical analysis, Paragraph 1): “We excluded invalid HIV self-test and weak HIV-positive self-test results from our FSW-interpreted HIV self-test sensitivity and specificity calculations because these testing outcomes are less common and we wanted to focus on how well FSWs could interpret clear self-test results.”

6. Please clearly explain how data was analyzed, you estimated binomial 95% CI for which variables. Did you use raw diagnostics (false positive, false negative, true positive, true negative) to calculate sensitivity/specificity? Suggest you to calculate concordance, otherwise please justify why not. You are not comparing HIVST results to a “gold standard” sensitivity/specificity are not appropriate measures.

We have extensively revised our statistical analysis section for clarity in response to reviewer comments. We now only estimate binomial 95% CIs for our measurements of FSW-interpreted HIV self-test sensitivity and specificity. We have clarified the reference that we used for our measurements of FSW-interpreted HIV self-test sensitivity and specificity and have justified why we think these measurements are appropriate for this study.

(Methods, Statistical analysis, Paragraph 1): “We calculated the proportion of participants that completed steps in the HIV self-testing process and the proportion of participants that had difficulties with each of these steps. To measure FSW-interpreted HIV self-test sensitivity and specificity, we calculated the proportion of participants that correctly interpreted the strong HIV-positive (two clear bands) and strong HIV-negative (one clear band) self-test results. The ‘correct interpretation’ of the self-test result was determined from the research assistant drawings of the self-test results, which were interpreted as indicated in the manufacturer’s instruction guide. We excluded invalid HIV self-test and weak HIV-positive self-test results from our FSW-interpreted HIV self-test sensitivity and specificity calculations because these testing outcomes are less common and we wanted to focus on how well participants could interpret clear self-test results. We did not compare FSW-interpreted HIV self-test results with laboratory blood-based HIV test results, because when interpreted correctly by trained users, oral HIV self-testing has >99% sensitivity and specificity.[31–33] Stata 13.1 (Stata Corporation, College Station, TX) was used to estimate binomial 95% confidence intervals for these measurements. We then calculated the proportion of participants that described their various perceptions of HIV self-testing, including the next steps following HIV self-test results and accuracy of the HIV self-tests.”

7. Please explain how observation was performed and how many research assistants were observing the FSW. Was the research assistant taking note during or after the process? Also please include this effect in the discussion or the limitations

We have discussed in more detail how research assistants observed participants as they used the HIV self-test. We have clarified that only one research assistant observed each participant as they HIV self-tested:

(Methods, Usability of HIV self-test, Paragraph 2): “Research assistants recorded the steps participants took when using the HIV self-testing on a standardized checklist, based on the OraQuick instructions. The checklist included: (1) removed buffer cap, (2) put buffer in buffer-stand, (3) swapped upper or lower gum, and (4) put test stick in buffer. Research assistants were instructed to indicate if participants had difficulties with any of these HIV self-testing steps. Additionally, research assistants recorded if participants read the HIV self-test instructions and when they read these instructions. Only one research assistant observed

each participant as they HIV self-tested. All used HIV self-tests were taken to a backroom (where no participants were present) by the study coordinator so that the results could process in private.”

In the discussion section, we discuss the limitation of having research assistants observe participants as they HIV self-test:

(Discussion, Paragraph 5): “Second, the FSWs in this study were observed using the HIV self-tests instead of testing in isolation, which may have either positively or negatively influenced study results depending on if this made FSWs more nervous or careful while self-testing or interpreting self-test results.[16]”

Results

1. How many participants were offered to participate and how many refused? Were there any lost participants?

We have clarified how many participants had the opportunity to participate in the study and how many refused. Since the study took place over the course of a few hours, no participants were lost, although not every participant completed each part of the study. We have now included this information in the manuscript.

(Results, Paragraph 1): “In October 2016, 118 FSWs in urban Kampala, Uganda completed day one of the two-day peer educator training. Of these, 104 (88%) agreed to perform an HIV self-test and 99 (84%) interpreted a HIV self-test and completed the questionnaire. Fourteen of the FSWs who participated in day one of the two-day peer educator training refused to perform HIV self-testing and five of the participants who performed HIV self-testing refused to interpret an HIV self-test result.”

2. Please do not repeat information in tables and figures. Figure 2 and table in appendix 1 have the same information.

We have cut Appendix Table 1 from the manuscript. None of the latest tables and figure repeat themselves.

3. Revise titles of figures and tables. The title should clearly indicate what results are shown in the table/figure. Table 1 for example -completed task: tasks accomplished in a defined period of time, the categories are mentioning steps for product use. Please revise CI95% calculation for tables, this statistical analysis does not correspond. CI95% are population parameters for which the difference between the parameter and the observed estimate is not statistically significant.

We have revised the titles of the figures and tables so that they more clearly indicate what the results are showing:

Table 1. Socio-demographic characteristics of study participants. N=104

Table 2. The percentage of participants that completed and had difficulties with steps in the HIV self-testing process. N=104*

Table 3. Participant perceptions of the next steps following HIV self-test results. N=99

Figure 1. Participants' interpretations of the HIV self-test results.

Figure 2. Participants' belief in the HIV self-test results, interest in HIV self-testing, and willingness to pay for HIV self-testing.

We no longer provide 95% confidence intervals for outcomes that report the proportion or participants that completed a task or reported an outcome.

4. Please provide more information about steps performance: did all the FSW read instructions fully or partially, difficulties removing buffer cap, buffer stand, swabbing their gums and putting the stick in the buffer. It's important to provide nuances about completeness and difficulties in the step. i.e. 77/99 were able to swab upper gum, but how many had difficulties doing so.

In the study, research participants indicated if participants were observed having difficulties with any of the particular steps in the HIV self-testing process. We have added these observations to our analysis.

(Methods, Usability of HIV self-test results, Paragraph 2): “Research assistants recorded the steps participants took when using the HIV self-testing on a standardized checklist, based on the OraQuick instructions. The checklist included: (1) removed buffer cap, (2) put buffer in buffer-stand, (3) swapped upper or lower gum, and (4) put test stick in buffer. Research assistants were instructed to indicate if participants had difficulties with any of these HIV self-testing steps. Additionally, research assistants recorded if participants read the HIV self-test instructions and when they read these instructions.”

(Results, Paragraph 3, Table 2): “Table 2 highlights the percentage of FSWs who completed and had difficulties with steps in the HIV self-testing process. While the majority of participants completed each of the individual steps (80% removed the buffer cap; 50% put the buffer in the buffer stand; 80% swabbed the upper or lower gum; 74% put the test stick in the buffer), only 33% of participants completed all four of these steps and the majority of participants (86%) were observed by research assistants as having difficulties with at least one of the steps. The majority of participants read the manufacturer’s self-test instructions prior to testing (58%) or during the HIV self-testing process (57%), but 49% were observed as having difficulties with the instructions.”

Table 2. The percentage of participants that completed and had difficulties with steps in the HIV self-testing process. N=104*

Step	Completed N (%)	Had difficulty N (%)
1. Removed buffer cap	80 (80%)	39 (38%)
2. Put buffer in buffer-stand	52 (50%)	72 (69%)
3. Swabbed upper or lower gum	82 (80%)	53 (51%)
4. Put test stick in buffer	77 (74%)	35 (34%)
All steps 1-4¹	34 (33%)	14 (13%)
Any step 1-4²	102 (98%)	89 (86%)
Read instructions prior to testing	61 (58%)	51 (49%) ³
Referenced instructions during testing	59 (57%)	

*Denominators vary based on completeness in reporting

¹Participants were observed completing all steps 1-4 or having difficulties with all steps 1-4.

²Participants were observed completing any of the steps 1-4 or having difficulties with any of the steps 1-4.

³Participants were observed having any difficulties with reading the instructions.

Reviewer: 3

Reviewer Name: Enrico Girardi

Institution and Country: National Institute for Infectious Diseases "L. Spallanzani" IRCCS, Rome, Italy

Please state any competing interests: None declared

This paper provides useful information for those planning the implementation of HIV self-testing programs. Some more details should be provided.

We thank the reviewer for this comment and have addressed their additional comments below.

Regarding the written information provided to participant, was it a mere translation from English to local language?

We have clarified that research assistants translated the informed consent from English to local languages, when necessary:

(Methods, Participants, Paragraph 1): “All participants provided written informed consent in the language in which they were most comfortable. Those who could not write provided consent with a thumbprint.”

For both the written and pictorial information, there was any adaption to local context and was understandability for local population pre-tested?

All participants received the written and pictorial information in both English and Luganda (the primary local language in Kampala). We used the instruction guides provided by OraQuick (available in both languages) and did not adapt these guides in any way.

(Methods, Usability of HIV self-tests, Paragraph 1): “All FSWs had access to written and pictorial OraQuick HIV self-test instructions (available in both English and the local language, Luganda), which included a guide on how to use the self-test as well as instructions on how to interpret the self-test results and link to care. We did not adapt this guide for use in this study.”

Is there any information on literacy/education of participants? Did they understand English?

We have added socio-demographic information of study participants to our results section and clarified that all public schooling on reading and writing in Uganda is conducted in English:

(Results, Paragraph 2): “Table 1 describes socio-demographic characteristics of study participants. The median age of study participants was 33 years (interquartile range: 29 to 27 years). The majority of participants (59%) reported completing some level of secondary education (9 to 12 years) and 90% of participants reported the ability to read and write. In Uganda, all public schooling on reading and writing is conducted in English. Almost all participants (83%) had worked as a peer educator at least once before participating in this study, 59% of which had previously worked for Uganda’s Most at Risk Population Initiative (also known as MARPI), which we used to help recruit peer educators for this study.”

Table 1. Socio-demographic characteristics of study participants.

N=104

Characteristics	N (%)
Age (median, IQR)	33 (29 to 37)

Education

No formal (0 years)	4 (4%)
Primary/Junior (<9 years)	28 (27%)
Secondary (9-12 years)	61 (59%)
Vocational	5 (5%)
Tertiary	6 (6%)
Can read and write	94 (90%)
Previously worked as a FSW peer educator	86 (83%)
Times works as a peer educator (median, IQR)	2 (1 to 4)

VERSION 2 – REVIEW

REVIEWER	Carmen Figueroa World Health Organization, Switzerland
REVIEW RETURNED	12-Jun-2018

GENERAL COMMENTS	Dear authors, Thank you very much for carefully revising your manuscript. I have some additional suggestion:  1. Please consider that an HIVST does not provide a definitive diagnosis, reactive results always require further testing and confirmation using validated national algorithm and non-reactive results will depend on the ongoing risk of HIV exposure. Therefore your approach to evaluate sensitivity and specificity from the research assistant drawing and considering sensitivity and specificity of Oraquick HIVST, is not appropriate. Sensitivity and specificity are used to evaluate the ability of a test to correctly identify presence of absence of a certain disease. Therefore you would need a reference gold standard for comparison, a reference that correctly identifies people. As this is not your case, you should be using concordance between FSW and research assistant, positive percent agreement/negative percent agreement or any other method used in the description of performance of diagnostic test where no true gold standard exists. 2. Please consider using the terminology “unassisted” HIVST, it is my opinion pre-test training is not clear enough; an in-person demonstration could be provided during HIVST. 3. Please state how many of the FSW having difficulties using the test were not able to complete HIVST, any particular key step 4. Please re-consider using “realistic scenario”, from one end you mention a “realistic scenario” providing HIVST only with instructions, and from another end you mention a “non-realistic scenario” where FSW cannot ask for help. Suggest you to only describe how HIVST was provided. 5. Literacy and intelligence are related to an extent, but being illiterate should not be the only explanation to have difficulties when using HIVST. Suggest to nuance your statements about no literacy/low literacy equals to difficulties/no capacity to use HIVST. 6. Please remove or clarify your statement about using HIVST to test others. HIVST can be offered to others (partners, friends, family members, not to children), by definition people should be testing themselves. 7. Suggest you to update the search of HIVST policy, a lot has happened since 8 Sept 2015, particularly in sub-Saharan African countries.
---

	8. Be consistent when referring to instructions for use, suggest you to avoid using guide. 9. Please note that capital N, refer to population size and n, refers to sample size. Suggest to use lower cap in all your tables, figures, and text, you are referring to sample size. 10. Not clear to me if invalid test were also distributed for interpretations. Please can you clarify? 11. Please specify if participants were only interpreting one possibility of result? How many kits per participant were distributed? If test were distributed randomly, how did you evaluate participants' ability to interpret all range of results (reactive, non-reactive, invalid)? If not, participants were then interpreting one possibility of result, this is another source of bias: you cannot guarantee ability to read reactive vs non-reactive vs invalid. 12. Please can you clarify what does the 0% in table 3 means: none of the participant answered this question or none of the participant expressed their intention to repeat testing in 12months? 13. Please read guidelines on how to prepare a table: labels of categories, what to do when you have low rate of reponse for several variables, etc. 14. Why are you not calculating statistical differences between HIV negative, HIV positive and invalid? 15. Please revise the strengths and limitations, it seems to me they are mixed. Suggest you to start with strengths. 16. I still find your findings do not fully support your conclusion: you're not reporting how many of the FSW having difficulties resulted in inability to complete the test or to obtain an invalid result. Looking at your findings despite the difficulties participants were able to perform the test. And you had a modest number of invalids 6%, normally the invalid rate for an RDT should not exceed 5%. Also please revise for clarity your final statement about evaluating support tools, RCT to determine levels of training..
--	--

REVIEWER	Enrico Girardi National Institute for Infectious Disease L. Spallanzani - IRCCS , Rpme , Italy
REVIEW RETURNED	04-Jun-2018

GENERAL COMMENTS	The issues raised in my review were satisfactorily addressed
--

VERSION 2 – AUTHOR RESPONSE

Reviewer: 2

Reviewer Name: Carmen Figueroa

Institution and Country: World Health Organization, Switzerland

Please state any competing interests: None declared

Dear authors,

Thank you very much for carefully revising your manuscript.

I have some additional suggestion:

1. Please consider that an HIVST does not provide a definitive diagnosis, reactive results always require further testing and confirmation using validated national algorithm and non-reactive results will depend on the ongoing risk of HIV exposure. Therefore your approach to evaluate sensitivity and specificity from the research assistant drawing and considering sensitivity and

specificity of Oraquick HIVST, is not appropriate. Sensitivity and specificity are used to evaluate the ability of a test to correctly identify presence or absence of a certain disease. Therefore you would need a reference gold standard for comparison, a reference that correctly identifies people. As this is not your case, you should be using concordance between FSW and research assistant, positive percent agreement/negative percent agreement or any other method used in the description of performance of diagnostic test where no true gold standard exists.

We no longer describe our estimates of how well participants could interpret strong HIV-positive (two clear bands) or strong HIV-negative (one clear band) self-test results as estimates of HIV self-testing sensitivity and specificity. We now describe these estimates as concordance between the participants and the manufacturer's interpretations of these self-test results. We have updated this description throughout the manuscript:

(Abstract): "We calculated the concordance between FSWs' interpretations of self-test results with those indicated in the manufacturer's instructions. ... The concordance between FSWs' interpretations of self-test results and that indicated in the instructions was 73% (95%CI 56-86%) for HIV-positive self-tests and 68% (95%CI 54-80%) for HIV-negative self-tests."

(Methods, Paragraph 9): "We also used proportions to calculate how participants interpreted HIV-positive, HIV-negative, and invalid HIV self-test results. We derived "correct" interpretations of self-test results using the self-test manufacturer instructions to interpret researcher assistant drawings of the result given to participants. We additionally calculated concordance between participants' interpretations and the manufacturer's interpretations of the strong HIV-positive (two distinct bands) and strong HIV-negative (one distinct band) self-test results. We excluded invalid HIV self-test and weak HIV-positive self-test results from our concordance measurements because these testing outcomes are less common and we wanted to focus on how well participants could interpret clear results. We did not compare FSW-interpreted HIV self-test results with laboratory blood-based HIV test results, because when interpreted correctly by trained users, oral HIV self-testing has >99% sensitivity and specificity.[31–33] We used Stata 13.1 (Stata Corporation, College Station, TX) to estimate binomial 95% confidence intervals for our concordance measurements."

(Results, Paragraph 5): "The concordance between participants' interpretations of self-test results and that indicated in the manufacturer's instructions was 73% (95%CI 56-86%) for HIV-positive self-tests and 68% (95%CI 54-80%) for HIV-negative self-tests."

(Discussion, Paragraph 1): "The concordance between FSWs' interpretations and the manufacturer's interpretations of strong HIV-positive and strong HIV-negative self-test results in this study is significantly lower than HIV self-test sensitivity and specificity measurements from other studies,[14–23] but consistent with that measured in another Kampala-based FSW population.[25]"

2. Please consider using the terminology "unassisted" HIVST, it is my opinion pre-test training is not clear enough; an in-person demonstration could be provided during HIVST.

We have modified our language throughout the text so that we no longer using the terminology "pre-test training". This has now been replaced with "assisted" and "unassisted" HIV self-testing throughout the manuscript.

3. Please state how many of the FSW having difficulties using the test were not able to complete HIVST, any particular key step.

We have added a sentence in our results section indicating how many of the FSW having difficulties using the HIV self-test were not able to complete HIVST, any particular key step:

(Results, Paragraph 3): “Of the 89 participants that had difficulties with at least one of the steps, 66% (n=69) were unable to complete at least one the four indicated steps in the self-testing process.”

4. Please re-consider using “realistic scenario”, from one end you mention a “realistic scenario” providing HIVST only with instructions, and from another end you mention a “non-realistic scenario” where FSW cannot ask for help. Suggest you to only describe how HIVST was provided.

We agree with the reviewer and have modified our use of “realistic scenario” throughout the text:

(Article summary): “This is one of few studies to explore unassisted oral HIV self-test performance among female sex workers, which represents a scenario where female sex workers might obtain HIV self-tests from pharmacies or friends.”

(Discussion, Paragraph 3): “Understanding unassisted HIV self-test performance is important because this reflects a scenario where FSWs may purchase an HIV self-test from a pharmacy or receive one from a friend.”

5. Literacy and intelligence are related to an extent, but being illiterate should not be the only explanation to have difficulties when using HIVST. Suggest to nuance your statements about no literacy/low literacy equals to difficulties/no capacity to use HIVST.

We agree with the reviewer and have taken out all discussion of low health literacy and education among FSWs as a potential explanation for their observed difficulties HIV self-testing.

6. Please remove or clarify your statement about using HIVST to test others. HIVST can be offered to others (partners, friends, family members, not to children), by definition people should be testing themselves.

At the suggestion of the reviewer, we have removed our statement about how HIV self-test might be used to test others outside of the controlled environment of the healthcare system.

(Introduction, Paragraph 2): “Understanding how well individuals can interpret HIV self-test results without bias of prior knowledge is important because as HIV self-testing moves HIV testing outside the unregulated environment of the health system,[8] individuals might use an HIV self-test for first time HIV testing.”

7. Suggest you to update the search of HIVST policy, a lot has happened since 8 Sept 2015, particularly in sub-Saharan African countries.

We apologize, that was an incorrect date in the reference section. We have checked the material posted on this website (HIST.org) and updated the citation with the current date:

(References): “HIVST.org. Policy & Regulations for HIVST. HIVST.org. <http://www.hivst.org/policy-regulations-for-hivst-1/> (accessed 26 Jun 2018).”

8. Be consistent when referring to instructions for use, suggest you to avoid using guide.

We have clarified our language around the OraQuick HIV self-testing instructions throughout the manuscript. At the suggestion of the reviewer, we no longer use the word “guide” when referencing the instructions:

(Methods, Paragraph 4): “All participants had access to written and pictorial OraQuick HIV self-test instructions (available in both English and the local language, Luganda), which included information on how to use the self-test as well as information on how to interpret the self-test results and link to care”

9. Please note that capital N, refer to population size and n, refers to sample size. Suggest to use lower cap in all your tables, figures, and text, you are referring to sample size.

We have updated our tables so that we only use a lower case “n” when referring to sample size.

10. Not clear to me if invalid test were also distributed for interpretations. Please can you clarify?

All types of HIV self-test results (e.g., HIV-positive, HIV-negative, and invalid) were distributed to participants for interpretation. We have clarified this in the methods section:

(Method, Paragraph 6): “Once the HIV self-test results had processed, these self-tests (including HIV-negative, HIV-positive, and invalid results) were placed in a large opaque bag by the study coordinator. Research assistants drew a random HIV self-test from the opaque bag for participants to interpret.”

11. Please specify if participants were only interpreting one possibility of result? How many kits per participant were distributed? If test were distributed randomly, how did you evaluate participants' ability to interpret all range of results (reactive, non-reactive, invalid)? If not, participants were then interpreting one possibility of result, this is another source of bias: you cannot guarantee ability to read reactive vs non-reactive vs invalid.

Participants only interpreted one randomly drawn HIV self-test. Since participants were randomly distributed an HIV self-test to interpret, we can measure participants' ability to interpret a range of results from the collective group's performance. In a real-life scenarios, individuals who HIV self-test will only see one HIV self-test at a time. If participants interpret many different HIV self-test results consecutively, their interpretation of later HIV self-test results may be biased by their interpretation of previous HIV self-test results. This is why we elected to have participants interpret one HIV self-test result in this study. We have clarified this in our methods section:

(Methods, Paragraph 6): “Participants only interpreted one randomly drawn HIV self-test instead of many so that previous interpretations of HIV self-test results would not bias later interpretations of HIV self-test results.”

12. Please can you clarify what does the 0% in table 3 means: none of the participant answered this question or none of the participant expressed their intention to repeat testing in 12 months?

The 0% in Table 3 means that none of the participants responded that they should test for HIV again in 12 months. We have clarified this with a footnote in Table 3 and have additionally only listed the n for these participants, not the percentage, in the table.

Table 3. Participant perceptions of the next steps following HIV self-test results. N=99

Next step	HIV self-test result		
	HIV-negative n (%)	HIV-positive n (%)	Invalid n (%)
Test again in 12 months ¹	0	0	0
Test again in 3 months	24 (25%)	3 (3%)	4 (5%)
Retake another oral HIV self-test	8 (8%)	7 (7%)	51 (58%)
Get confirmatory test at clinic	20 (21%)	45 (46%)	32 (36%)
Don't know	2 (2%)	4 (4%)	3 (3%)
Other: Get counseling	-	21 (21%)	1 (1%)
Other: Start treatment	1 (1%)	41 (42%)	2 (2%)
Other: Use protection	53 (53%)	4 (4%)	3 (3%)
Other: Look after yourself	1 (1%)	-	-
Other: Suicidal thoughts	-	1 (1%)	-
Other: Test again in 1.5 months	-	1 (1%)	-
Other: Consult others	-	-	2 (2%)
Other: Give up on testing	-	-	1 (1%)

Abbreviations: sample size (n); participants did not mention this “other” category (-)

¹None of the participants responded that they should test for HIV again in 12 months, as indicated with n=0

13. Please read guidelines on how to prepare a table: labels of categories, what to do when you have low rate of response for several variables, etc.

We have carefully reviewed our tables and ensured that they are presented in accordance with BMJ Open guidelines.

14. Why are you not calculating statistical differences between HIV negative, HIV positive and invalid?

We are not calculating statistical differences between HIV negative, HIV positive, and invalid because participants only interpreted one HIV self-test result. Additionally we did not interpret statistical differences in their perceptions of the next steps following various HIV self-test results because the responses to these questions were varied and not directly comparable.

15. Please revise the strengths and limitations, it seems to me they are mixed. Suggest you to start with strengths.

We have revised the strengths and limitations section of this paper, starting with strengths.

(Discussion, Paragraph 5): “This study had a number of strengths and limitations. An important strength of this study is that participants interpreted an HIV self-test result that was not their own and thus interpretations of self-test results were not biased by knowledge of previous HIV test results and past HIV risk encounters. Another strength was that participants performed unassisted HIV self-testing, which enables us to understand HIV self-testing performance among FSWs in environments where assistance might be unavailable. Limitations of this study, however, included sample size, which was small because this study was a sub-study of a larger HIV self-testing randomized controlled trial.[8] In this study participants were also observed using the HIV self-tests instead of testing in isolation, which may have either positively or negatively influenced study results depending on if this made participants more nervous or careful while self-testing or interpreting self-test results.[16] While unlikely, research assistants may have incorrectly drawn the results of random HIV self-tests given to participants for interpretation, potentially resulting in a downward bias of our FSW-interpreted HIV self-test sensitivity and specificity estimates. Some of the self-tests interpreted by study participants may have been interpreted 40 minutes after use, which is outside of the recommended interpretation period. This may have resulted in a greater number of weak HIV-positive self-test results, which are more difficult to interpret. For this reason, weak HIV-positive self-test results were excluded from our measurement of FSW-interpreted HIV self-test sensitivity. Finally, some participants may have been living with HIV and on antiretroviral treatment, which may have biased their perceptions of the next steps following an HIV-positive self-test result and likely resulted in an overestimation of participants reporting that individuals who self-test HIV-positive should start treatment.”

16. I still find your findings do not fully support your conclusion: you're not reporting how many of the FSW having difficulties resulted in inability to complete the test or to obtain an invalid result. Looking at your findings despite the difficulties participants were able to perform the test. And you had a modest number of invalids 6%, normally the invalid rate for an RDT should not exceed 5%. Also please revise for clarity your final statement about evaluating support tools, RCT to determine levels of training...

We have revised our conclusions so that they specifically relate to what we measured in this study: FSWs' ability to complete the HIV self-testing process and FSWs' ability to correctly interpret HIV self-test results when unassisted:

(Conclusions, Paragraph 1): “This study demonstrated that FSWs in urban Uganda have difficulties with the HIV self-testing process and have difficulties interpreting HIV self-test results when unassisted.”

We have also revised our final statement, excluding the recommendation to conduct RCTs to determine the appropriate level of HIV self-testing assistance:

(Conclusion, Paragraph 1): “Simpler HIV self-testing support tools (i.e., instructions or demonstration videos), redesign of HIV self-test, or the use of HIV self-testing counselors should be explored and additional research should be conducted to determine the appropriate level of HIV self-testing assistance.”

Reviewer: 3

Reviewer Name: Enrico Girardi

Institution and Country: National Institute for Infectious Disease L. Spallanzani - IRCCS , Rome , Italy

Please state any competing interests: None declared

The issues raised in my review were satisfactorily addressed.

We thank the reviewer for their comment.

VERSION 3 – REVIEW

REVIEWER	Carmen Figueroa World Health Organization, Switzerland
REVIEW RETURNED	19-Jul-2018
GENERAL COMMENTS	Dear authors Thanks for carefully considering my suggestions. All my comments have been addressed